# Three-Dimensional Evaluation of the Frontal Sinus in Koreans

**DOI:** 10.3390/ijerph19159605

**Published:** 2022-08-04

**Authors:** Jeong-Hyun Lee, Jong-Tae Park

**Affiliations:** 1Department of Oral Anatomy, Dental College Dankook Institute for Future Science and Emerging Convergence, Dankook University, Cheonan 330-714, Korea; 2Department of Bio Health Convergency Open Sharing System, Dankook University, Cheonan 330-714, Korea

**Keywords:** 3D, frontal sinus, CBCT, cranial bone, supraorbital line

## Abstract

(1) Background: Among the four paranasal sinuses, the frontal sinus is in the frontal bone. Recent research trends have been focusing on identifying sex based on the frontal sinus. Thus, this study aimed to provide reference data for the frontal sinus in Korean adults by comparing their sizes using a 3D program. Moreover, this study examined the correlation between the size of the frontal sinus and the length of cranial bone. (2) Methods: Cone-beam computed tomography (CBCT) data were obtained from 60 (male 30, female 30) patients in their 20 s who visited the Department of Dankook University Hospital (DKUDH IRB 2020-01-007). The provided patient CBCT data were utilized to reconstruct the patients’ frontal sinuses and cranial bones in 3D using the Mimics (version 22.0, Materialise, Leuven, Belgium) 3D program. All measurements were analyzed using SPSS (ver. 23.0, IBM Corporation, Armonk, NY, USA). (3) Results: By comparing the frontal sinus size of Korean adults according to sex using a 3D program, this study revealed that males had larger frontal sinuses than females. (4) Conclusions: The findings of this study could help in preventing complications that occur in various clinical treatments and analyzing the growth of the frontal sinus in the future.

## 1. Introduction

There are four paranasal sinuses, with the frontal sinus located in the frontal bone [1]. The frontal sinus begins to develop at 2–3 years of age, can be observed on radiographs at 5–6 years of age, and is fully formed by 20 years of age [2]. The frontal sinus is also divided into its right and left parts by the frontal septum, and its size [3] and shape vary with race and climate [4]. There are therefore individual differences in the shape of the frontal sinus and surrounding anatomy [3]. Studies have recently been conducted on sex identification based on the frontal sinus [1,2,3,4,5,6,7,8,9,10,11,12,13,14,15,16,17,18,19,20,21,22,23,24,25].

Sinusitis is the most common frontal sinus disease [4], caused by various factors, such as bacteria, fungi, and myxoma, and may lead to symptoms, including facial pain, headache, nasal stiffness, and vision loss [3]. As the frontal sinus is adjacent to the ethmoid cavity, surgical management has been developed [5] and has been increasingly used for the treatment of sinusitis in the last few years [6].

Frontal bone fractures account for about 5% of facial traumas [7] and can cause frontal bone retraction as well as injuries to the dura mater and supraorbital foramen [8,9]. Most studies therefore tend to classify the frontal bone fractures and identify the location of the frontal sinus [1,2,3,4,5,6,7,8,9,10,11,12]. Buller et al. [10] found that the types of frontal bone fractures varied with the shape of the frontal sinus, and Ruiz et al. [11] found a correlation between the size of the frontal sinus and the cranial bone width. While these results come from ongoing research into the shape and growth of the frontal sinus [12], most of these studies have focused on the measurement of the facial bones of dead bodies and 2D measurements on radiographs [1,2,3,4,5,6,7,8,9,10,11,12,13,14,15,16,17,18,19,20,21,22,23,24,25], which may be inaccurate for measuring the 3D human body [14].

This study therefore used a 3D program to compare the size of the frontal sinus between the sexes and malocclusion in Korean adult subjects in order to provide reference data. It also aimed to determine the correlation between the size of the frontal sinus and the cranial bone length.

## 2. Materials and Methods

### 2.1. Subjects

CBCT data were obtained from 60 subjects (30 males, 30 females) aged 20–29 years with malocclusions who visited the Dankook University Dental Hospital. Patients without common or bilateral stones in the frontal sinus were selected for evaluation of the frontal sinus. They did not have any missing teeth, asymmetries, or systemic disease. The required number of subjects was calculated using the G*Power 3.1 program (HHU, Dusseldorf, Germany). The required number of total participants was calculated with two-tails, effect size (d) of 0.8, α err prob of 0.05, and power (1-β err prob) of 0.8 [26]. A total of 50 patients was required with 26 patients in each group, and 10 additional patients were added for increased robustness.

Since the CBCT data in this study were analyzed retrospectively, exemption from obtaining informed consent was requested, and the study was conducted after receiving approval from the IRB of Dankook University Dental Hospital (DKUDH IRB 2020-01-007).

### 2.2. CBCT Data

All images were acquired by the same experienced technician. To reduce individual differences in the size of the frontal sinus between the participants, a Frankfort Horizontal (FH) plane was set perpendicular to the floor. Further, the sagittal midline of the face of each patient was matched with a computed tomography (CT) equipment (Alphard 3030, Asahi, Kyoto, Japan) for imaging of the cranial bone. The imaging conditions were as follows: 0° gantry angle, 120 kV, and auto mA. CBCT scanning was conducted using the following parameters: slice increment of 0.39 mm, slice thickness of 0.39 mm, Slice Pitch, 3; Scanning time, 4 s; voxel size, 0.39 mm × 0.39 mm × 0.39 mm and matrix size of 512 pixels by 512 pixels. The CBCT data were provided in DICOM format.

### 2.3. 3D Modeling

The Mimics 3D program (version 22.0, Materialise, Leuven, Belgium) was used to construct a 3D model of the frontal sinus of each subject in the coronal, sagittal, and axial views based on the produced DICOM files. The frontal sinus and cranial bone in three dimensions were extracted by adjusting the Hounsfield unit (HU) grayscale values [27]. The Hounsfield criteria were also configured according to the specified average range of the Mimics software to observe the bone and soft tissues (Figure 1).

#### 2.3.1. Frontal Sinus 3D Modeling

Masking was performed using minimum value of −1024 HU and a maximum value of −302 HU to reconstruct the frontal sinus cavity in three dimensions. Soft tissues other than the frontal sinus were obtained using Boolean operations. The frontal sinus was then separated into its left and right parts using the edit-mask function, and the right and left mask were constructed (Figure 2).

#### 2.3.2. Cranial Bone 3D Modeling

To measure the cranial bone transversal width, masking was performed using minimum value of −277 HU and maximum value of 3071 HU. Unnecessary soft tissues were removed using the edit-mask function, and then masking was performed. The data collected were extracted using the calculate-part function to convert it to an STL file. Each item was then measured using the distance function.

### 2.4. Measurements

All measurements were made after adjusting the FH line horizontally. Moreover, as the lower part of the frontal sinus is adjacent to the ethmoid sinus, the nasofrontal suture was used as the reference line to accurately measure the upper part of the frontal sinus. The maximum width of the frontal sinus (XW) was measured from the nasal septum, and maximum frontal sinus height (XH) was measured from the nasofrontal suture. Frontal sinus width (ZW) measured the part with the longest anteroposterior length.

All measurements were made after measuring Lee and Park twice each, and the mean was calculated for evaluation. The reliability of the measured data was also analyzed, which produced a Cronbach’s α of 0.618. Measurement items are listed in Table 1. The cranial bone width measurements are listed in Table 2 (Figure 3 and Figure 4).

### 2.5. Statistical Analysis

Measurements were analyzed using SPSS (version 23.0, IBM Corporation, Armonk, NY, USA). As the sample size was small in this study, the Shapiro–Wilk test was performed for the normality test. Skewness and kurtosis were 0.427 and 0.833, respectively. Thus, the Mann–Whitney test, a non-parametric test, was performed to compare the frontal sinus size according to sex. The Kruskal–Wallis test, a non-parametric test, was performed to compare the frontal sinus size according to malocclusion. Pearson’s product-moment correlation was performed to assess the correlation between the frontal sinus and cranial bone width, and linear regression analysis was conducted to assess the effects of the cranial bone width on the frontal sinus. All statistical analyses were conducted with a 95% confidence interval, and a *p*-value less than 0.05 was considered statistically significant.

## 3. Results

The results of the frontal sinus size comparison by sex are listed in Table 3 (Figure 5, Figure 6 and Figure 7).

### 3.1. Frontal Sinus Width (XW) and Height (XH)

The right frontal sinus width (RXW) was larger in males than in females, at 28.29 mm and 26.30 mm, respectively (<0.05), as was the left frontal sinus width (LXW), at 30.49 mm and 25.41 mm, respectively (<0.05). The right frontal sinus height (RXH) was larger in males than in females, at 30.09 mm and 26.31 mm, respectively (<0.05), as was the left frontal sinus height (LXH), at 30.93 mm and 27.42 mm, respectively (<0.05).

### 3.2. Frontal Sinus Width (ZW) and Volume (FSV)

The frontal sinus width (ZW) was larger in males than in females, at 25.22 mm and 19.69 mm, respectively (<0.05). The frontal sinus volume (FSV) was also larger in males than in females, at 9602 mm^3^ and 5611 mm^3^, respectively (<0.05).

### 3.3. Cranial Bone Transversal Width

The supraorbital line (SOL) was longer in males than in females, at 52.83 mm and 48.08 mm, respectively (<0.05). The cranial bone transversal width (CW) was also larger in males than in females, at 151.12 mm and 145.99 mm, respectively (<0.05).

The results of comparing the correlation between frontal sinus size and cranial bone width were as follows (Table 4).

### 3.4. Correlation between Frontal Sinus Width and Cranial Bone Transversal Width

Correlations were found among RXH, FSV, SOL, and CW, and among LXH, ZW, FSV, SOL, and CW. ZW was correlated with LXW, FSV, SOL, and CW in a sagittal view. These findings suggest that there is a significant correlation between the widths of the frontal sinus and the cranial bone.

### 3.5. Correlation between Frontal Sinus Height and Cranial Bone Transversal Width

Correlations were also found among RXW, FSV, SOL, and CW, and between LXW and FSV. These findings suggest that there is a significant correlation between the height of the frontal sinus and the cranial bone width.

### 3.6. Correlation between Frontal Sinus Volume and Cranial Bone Transversal Width

Correlations were found between FSV and RXW, RXH, LXW, LXH, ZW, SOL, and CW. These findings suggest that there is a significant correlation between FSV and the frontal sinus width, height, and cranial bone width.

The effects of the cranial bone width on the frontal sinus size are described in Table 5. The regression model analysis showed a satisfactory goodness-of-fit with F = 2.931 (*p* < 0.05) and had an explanatory power of 28% with R2 = 0.283. RXW and RXH had β = (0.058) and β = (0.161), respectively, suggesting that the cranial bone width did not affect the right frontal sinus size (*p* > 0.05). Similarly, LXW and LXH had β = 0.074 and β = −0.251, respectively, showing that the cranial bone width had no effects on the left frontal sinus size (*p* > 0.05). ZW had β = 0.113 while volume and SOL showed β = 0.174 and β = 0.240, respectively, suggesting that the cranial bone width did not affect the frontal sinus size (*p* > 0.05). Hence, β (+) was calculated for RXW, RXH, LXW, ZW, volume, and SOL, indicating that the cranial bone width did not have significant effects on the size of the frontal sinus.

The results of the frontal sinus size comparison by malocclusion are listed in Table 6.

### 3.7. Frontal Sinus Width (XW) and Height (XH) According to Malocclusion

The right frontal sinus width (RXW) was larger in Class I than in Class II and Class III, at 27.83 mm, 25.56 mm and 27.00 mm, respectively (>0.05), as was the left frontal sinus width (LXW), at 29.32 mm, 25.66 mm and 28.87 mm, respectively (>0.05). The right frontal sinus height (RXH) was larger in Class I than in Class II and Class III, at 29.07 mm, 28.28 mm and 27.26 mm, respectively (>0.05). However, the left frontal sinus height (LXH) was larger in Class II than in Class I and Class III, at 29.45 mm, 28.81 mm and 29.26 mm, respectively (>0.05).

### 3.8. Frontal Sinus Width (ZW) and Volume (FSV) According to Malocclusion

The frontal sinus width (ZW) was larger in Class I than in Class II and Class III, at 23.48 mm, 23.22 mm, and 20.68 mm, respectively (>0.05). The frontal sinus volume (FSV) was also larger in Class I than in Class II and Class III, at 8068 mm^3^, 7664 mm^3^, and 7088 mm^3^, respectively (<0.05).

### 3.9. Cranial Bone Transversal Width According to Malocclusion

The supraorbital line (SOL) was longer in Class I than in Class II and Class III, at 51.91 mm, 51.47, and 47.98 mm, respectively (>0.05). The cranial bone transversal width (CW) was larger in Class III than in Class I and Class II, at 149.82 mm, 148.89 mm, and 146.95 mm, respectively (>0.05).

## 4. Discussion

The frontal sinus is one of the paranasal sinuses of the cranial bone [13], and its shape varies for each individual [14]. As the frontal sinus is formed after adulthood by craniofacial bone pneumatization and hormones [15], studies are actively investigating the relationship between the cranial bone and frontal sinus [1,2,3,4,5,6,7,8,9,10,11,12,13,14,15,16,17,18,19,20,21,22,23,24,25]. Endoscopic sinus surgery (ESS) is performed to treat sinusitis and cranial bone diseases [16]. However, this surgery has a 0–1.5% chance of postoperative complications [17].

Most studies on the cranial bone and frontal sinus use linear measurements in two dimensions, and so the measured values might not be accurate [12]. There are also no studies on the reference values of frontal sinus dimensions in Koreans [1,2,3,4,5,6,7,8,9,10,11,12,13,14,15,16,17,18,19,20,21,22,23,24,25]. This study therefore used a 3D program to compare the size of the frontal sinus by sex, malocclusion, and to investigate the correlation between frontal sinus size and cranial bone length.

This study found that the frontal sinus was larger in males than in females, which is similar to previous studies that have compared the sizes of the frontal sinus [14].

Tehranchi et al. [18] found that males had larger frontal sinuses than females, but with no significant difference. In contrast, a significant difference in frontal sinus width according to sex was observed in the present study (*p* < 0.05). This discrepancy might be attributable to differences in the frontal sinus according to age. Yun et al. [12] found that the frontal sinus grows rapidly until the ages of 13–16 years. It is therefore expected that there could be age differences between this study, which evaluated subjects aged 20–29 years, and the study of Tehranchi et al. [18], which evaluated 12-year-old subjects. It therefore seems that caution is required when performing frontal sinus surgery depending on the age of the patient. Moreover, the evaluation of the frontal sinus size can aid in the endoscopic approach during ESS.

This study found that the frontal sinus was higher on both sides in males than in females (*p* < 0.05). However, Buyuk et al. [19] found no significant difference in the height of the frontal sinus according to sex, while Soman et al. [20] found that the right frontal sinus was higher in females than in males. This suggests that the height of the frontal sinus varies depending on the race and climate [21]. The differences between the studies of Buyuk et al. [19] and Soman et al. [20], which were conducted on Asians and Caucasians, respectively, might therefore have been expected. Caution is required when performing surgery related to the height of the frontal sinus depending on the race of the patient. In addition, osteoplasty through evaluation of the frontal sinus height may minimize the use of an external approach to the frontal sinus and prevent complications.

In this study, the mean FSV was 9602 mm^3^ in males and 5611 mm^3^ in females. However, Yun et al. [12] measured values of 8540 mm^3^ and 8230 mm^3^, respectively, in Korean subjects. However, the FSV evaluation of Pirner et al. [22] found that both sides were larger in males than in females, with mean values of 10,200 mm^3^ and 5500 mm^3^, respectively, which were similar to those in the present study. The differences in the mean FSV between the subjects are believed to be due to the continuous growth of the frontal sinus up to 20–29 years old [23]. Caution is therefore required to prevent side effects when performing adjacent frontal sinus surgery for adolescents in the growth phase.

When comparing the correlation between the size of the frontal sinus and the cranial bone width, this study found a correlation between the size of the frontal sinus and SOL, FSV, and cranial bone width. Ruiz et al. [11] found a correlation between the area of the frontal sinus and the cranial bone length, which was similar to the finding in the present study. However, our findings showed that the cranial bone width did not affect the frontal sinus size. In contrast, Blaney [24] showed that the cranial bone had significant effects on the frontal sinus shape. Those results may have been affected by the final formation of the frontal sinus, determined by the composition of craniofacial bones and hormones after adulthood [15]. Thus, further studies would be necessary to present more evidence.

Finally, as a result of comparing the forehead cave according to the malocclusion, it was found that the forehead cave was generally larger in the Class I group. However, in the research of Sabharwal et al. [25], it was found to be large in the Class III group, unlike the results of this research. The reason for the different experimental results is that in the research of Sabharwal et al. [25], the number of subjects in each group was not equal. Therefore, further research on malocclusion is needed in the future.

## 5. Conclusions

This study compared the frontal sinus size according to sex and malocclusion in Korean adults. As a result, the frontal sinus was found to be larger in men than in women. The frontal sinus was found to be larger in Classs I than in Class II and Class III. In addition, an analysis of the correlation between the frontal sinus size and cranial bone width showed that the frontal sinus size was correlated with SOL, volume, and cranial bone width. However, the cranial bone length did not affect the frontal sinus size. Our findings can help to prevent potential complications during clinical operations, such as facial fracture surgeries, and further studies are warranted in the future to confirm our findings.

## Figures and Tables

**Figure 1 ijerph-19-09605-f001:**
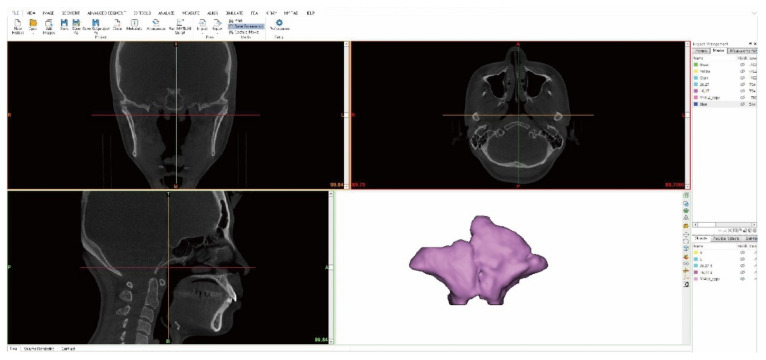
Mimics (version 22.0, Materialise, Leuven, Belgium).

**Figure 2 ijerph-19-09605-f002:**
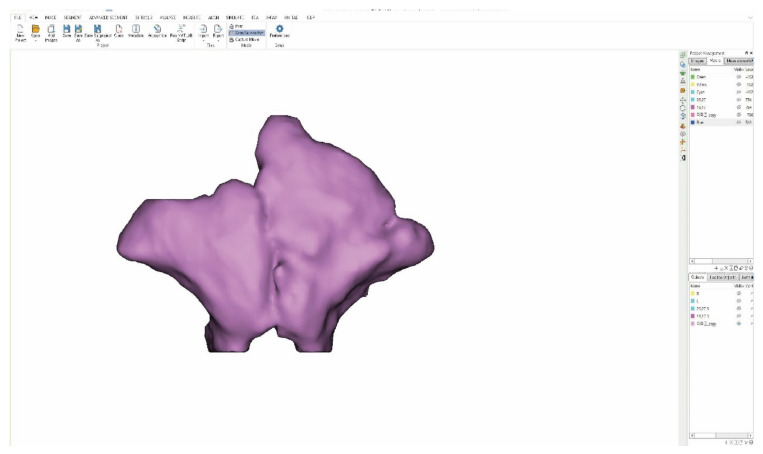
Frontal sinus 3D.

**Figure 3 ijerph-19-09605-f003:**
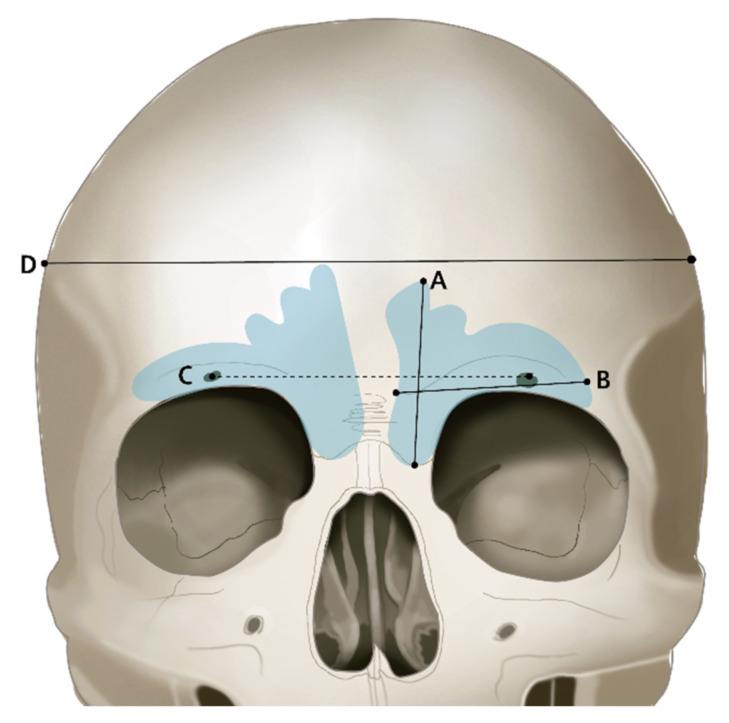
Coronal view. (A) Frontal sinus width (XW): Frontal sinus width in a coronal view; (B) Frontal sinus width (XH): Frontal sinus height in a coronal view; (C) Supraorbital line (SOL): Distance between supraorbital foramens; (D) Cranial bone transversal width: Full width of cranial bone in a coronal view.

**Figure 4 ijerph-19-09605-f004:**
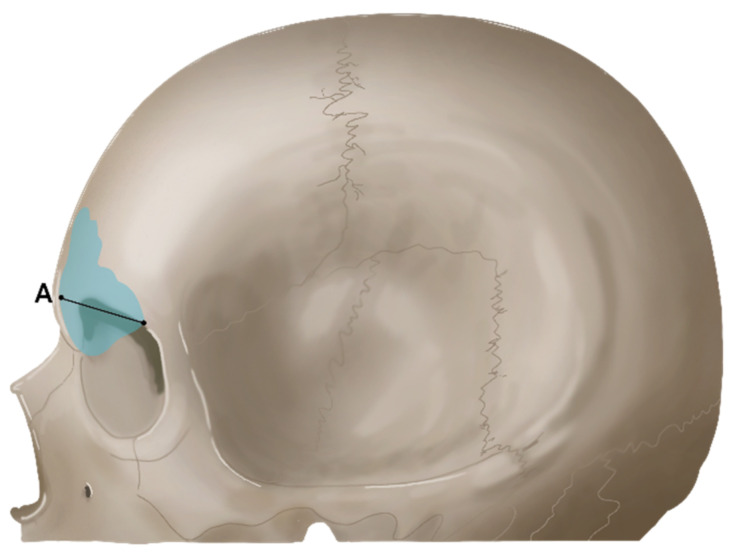
Sagittal view. (A) Frontal sinus width (ZW): Frontal sinus width in a sagittal view.

**Figure 5 ijerph-19-09605-f005:**
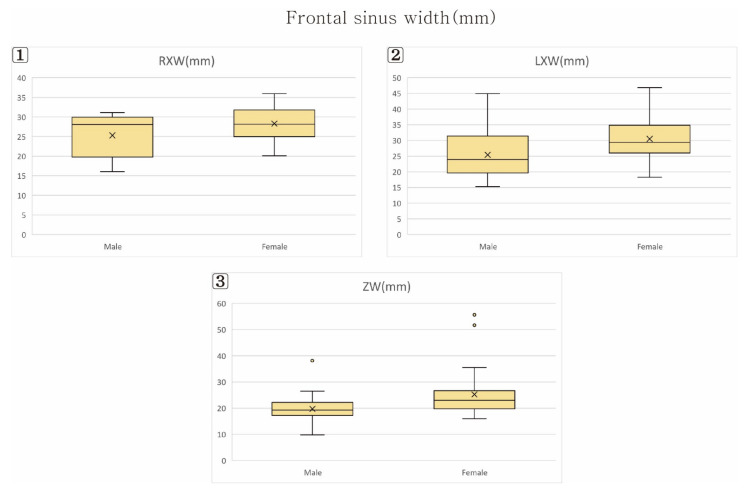
Frontal sinus width. (**1**) Right Frontal sinus width (RXW); (**2**) Left Frontal sinus width (LXW); (**3**) Frontal sinus width (ZW).

**Figure 6 ijerph-19-09605-f006:**
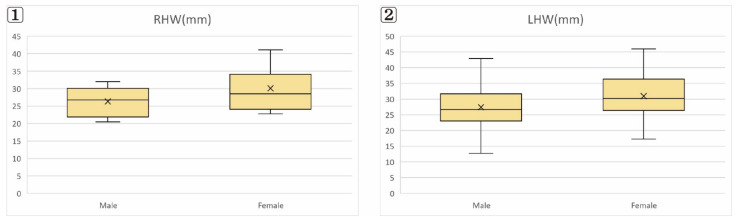
Frontal sinus height. (**1**) Right Frontal sinus height (RXH); (**2**) Left Frontal sinus height (LXH).

**Figure 7 ijerph-19-09605-f007:**
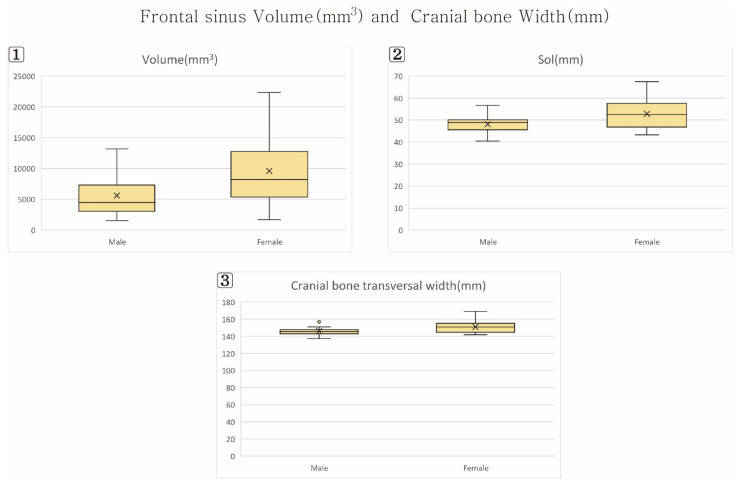
Frontal sinus Volume and Cranial bone width. (**1**) Frontal sinus Volume; (**2**) Supraorbital line (Sol); (**3**) Cranial bone transversal width (CW).

**Table 1 ijerph-19-09605-t001:** Frontal sinus measurements.

Parameter	Definition
Right frontal sinus width (RXW)	Width of right frontal sinus in a coronal view
Left frontal sinus width (LXW)	Width of left frontal sinus in a coronal view
Right frontal sinus height (RXH)	Height of right frontal sinus in a coronal view
Left frontal sinus height (LXH)	Height of left frontal sinus in a coronal view
Frontal sinus width (ZW)	Frontal sinus width in a sagittal view
Frontal sinus volume (FSV)	Frontal sinus total volume

**Table 2 ijerph-19-09605-t002:** Cranial bone width measurements.

Parameter	Definition
Supraorbital line (SOL)	Distance between supraorbital foramens
Cranial bone transversal width (CW)	Full width of cranial bone in a coronal view

**Table 3 ijerph-19-09605-t003:** Frontal sinus size comparison by sex.

Measurements	Male (N = 30)	Female (N = 30)	*p* Value
RXW (mm)	28.29 (4.76)	26.30 (5.34)	0.042 *
RXH (mm)	30.09 (6.03)	26.31 (3.84)	0.050 *
LXW (mm)	30.49 (7.15)	25.41 (7.58)	0.008 *
LXH (mm)	30.93 (6.69)	27.42 (7.17)	0.048 *
ZW (mm)	25.22 (9.08)	19.69 (5.25)	0.004 *
Volume (mm^3^)	9602 (5658)	5611 (3452)	0.002 *
SOL (mm)	52.83 (6.21)	48.08 (3.34)	0.001 *
Cranial bone transversal width (mm)	151.12 (6.49)	145.99 (4.54)	0.002 *

Data are mean (standard-deviation values); *p*-value were obtained by Mann-Whitey U test; * *p* < 0.05.

**Table 4 ijerph-19-09605-t004:** Correlation between frontal sinus size and cranial bone width.

	RXW (mm)	RXH (mm)	LXW (mm)	LXH (mm)	ZW (mm)	Volume (mm^3^)	SOL (mm)	Cranial Bone Transversal Width (mm)
RXW (mm)	1							
RXH (mm)	0.643 **	1						
LXW (mm)	NS	NS	1					
LXH (mm)	NS	NS	0.474 **	1				
ZW (mm)	0.230	0.228	0.335 **	0.158	1			
Volume (mm^3^)	0.705 **	0.710 **	0.669 **	0.480 **	0.341 **	1		
SOL (mm)	0.368 **	0.309 **	0.376 **	0.188	0.339 **	0.450 **	1	
Cranial bone transversal width (mm)	0.293 *	0.333 **	0.298 *	0.055	0.289 *	0.404 **	0.421 **	1

Significance levels, * *p* < 0.05 ** *p* < 0.01; NS, not significant.

**Table 5 ijerph-19-09605-t005:** Effect of cranial bone transversal width on frontal sinus size.

Measurements	B	SE	*β*	T (*p*)	F (*p*)	R2
Constant	129.168	8.477		15.238	2.931	0.283
RXW (mm)	0.068	0.225	0.058	0.301		
RXH (mm)	0.184	0.215	0.161	0.856		
LXW (mm)	0.058	0.138	0.074	0.422		
LXH (mm)	−0.216	0.136	−0.251	−1.590		
ZW (mm)	0.088	0.101	0.113	0.870		
Volume (mm^3^)	0.021	0.026	0.174	0.795		
SOL (mm)	0.267	0.154	0.240	1.741		

*p*-value were obtained by Simple linear regression.

**Table 6 ijerph-19-09605-t006:** Frontal sinus size comparison by malocclusion.

Measurements	Class I (N = 20)	Class II (N = 20)	Class III (N = 20)	*p* Value
RXW (mm)	27.83 (5.46)	25.56 (5.11)	27.00 (5.14)	0.976
RXH (mm)	29.07 (5.75)	28.28 (5.58)	27.26 (4.84)	0.430
LXW (mm)	29.32 (9.54)	25.66 (6.39)	28.87 (6.77)	0.591
LXH (mm)	28.81 (7.13)	29.45 (7.23)	29.26 (7.29)	0.294
ZW (mm)	23.48 (5.85)	23.22 (11.27)	20.68 (5.07)	0.901
Volume (mm^3^)	8068 (5378.99)	7664 (5144.70)	7088 (4868.17)	0.004 *
SOL (mm)	51.91 (5.38)	51.47 (4.14)	47.98 (6.15)	0.479
Cranial bone trans-versal Width (mm)	148.89 (5.42)	146.95 (5.00)	149.82 (7.58)	0.287

Data are mean (standard-deviation values); *p*-value were obtained by Kruskal-Wallis test; * *p* < 0.05.

## Data Availability

Original data are available upon request to the corresponding author.

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
