# Peer review of "Three-Dimensional Evaluation of the Frontal Sinus in Koreans"

_ijerph, 2022, doi:10.3390/ijerph19159605_

Round 1
Reviewer 1 Report
The authors present a relatively straight forward study on the volumetric and linear dimensions of the frontal sinus among a Korean population. These data add to the ever growing database of frontal sinus dimensions across varying groups. There are a number of these types of studies, so it isn’t quite clear why another one is needed. The authors mention that there aren’t many studies evaluating the frontal sinus using 3D methods in Korean populations, but that’s not really the case. Indeed, the authors already cite several of those studies. What the authors could do to set this study apart, however, is provide a different analytical method to the data and refocus their discussion on health-based implications. First, I recommend they run a regression analysis between the frontal sinus dimensions and cranial dimensions, instead of just a simple correlation. This would help this study stand out a bit more. Second, as written, the study also does not quite fit in the special issue it was submitted to. However, I think the authors could re-focus the discussion and make it more suitable. I provide detail comments on how to enhance the study based on each of the sections below.
Materials and Methods
The authors do a nice job of describing how the scans were taken, and the programs used. This is often not provided, so it’s nice to see it clearly represented here. However, I have questions regarding how were individuals oriented in the CBCT scanner (OR how were they oriented in the software)? For example the frontal sinus and its linear dimensions may present differently if individuals are in Frankfort horizontal position vs a Caldwell view or other typical radiographic orientation.
This need for detail on orientation ties in to needing more details on how measures were taken for repeatability. The authors have nice illustrations (very well done!), however the descriptions of how the frontal sinus was measured is still lacking. The authors mention “Measurements were made based on the highest point” (Line 95). I understand how the highest point makes sense for height, but I’m not sure how this relates to breadth. Frontal sinus breadth, in particular, is taken many different ways in the literature. Line B in Figure 3 shows breadth is taken from the intersinus septum to the most lateral edge of the sinus, but from which specific point on the septum and at what angle? For example, some studies would take breadth parallel to the supraorbital line, which doesn’t seem to be the case here. Was height simply from the most superior point to most inferior point (i.e. maximum height, regardless of angle to midline?)
The authors make no mention of absent sinuses, so I’m assuming none of their sample had bilaterally or unilateral asymmetry in the sinuses. This may be nice to explicit state if all sinuses were bilaterally present in the sample. If not, what was the frequency for unilateral absence, bilateral absence etc.
Finally, the authors should be more clear in regard to what they are referring to with the “cranial bone”. Cranial width (across the neurocranium) makes sense, but there isn’t a specific cranial bone in the skull. For example, when the authors say they took cranial bone thickness,
Results:
Did the authors use MannWhitney U tests because the data were not normally distributed or skewed? Results on normality testing and skewness should be provided so readers know appropriate statistics were used. It is also customary to provide the actual p-values, versus just p<0.05 etc.
There doesn’t seem to be mention of which correlation tests the authors conducted. Regardless, I think the correlation tests don’t actually tell you that much of the relationship. I would advise the authors replace the simple correlations with regression analyses. In the case with higher correlated variables (frontal sinus breadth to cranial breadth) based on the regressions, it would be beneficial to have a figure displaying a bivariate plot showing that relationship. Having the data points labeled for male vs female would further accentuated the claims the authors make regarding potential sex-based differences. Assuming these sex-based differences remain in the regression analyses, I would expect to see a somewhat clear differentiation with how the male versus females fall in the plot relative to the regression line.
Figure 5. The bar charts aren’t really effective at showcasing the differences in means with the variation between males and females. These should be replaced by boxplots.
Discussion
The authors should remove the first few sentences of discussion (Lines 60-163), I think this is just left over from the MDPI template.
Overall, the authors should work to rewrite the discussion to make this submission more suitable for a special issue on health. I would remove the discussion on forensic aspects for this special issue (I would limit it in the introduction as well). But, it would be helpful to add about the surgical or clinical implications. For example, how to larger/taller sinuses actually pose a hindrance to specific surgical interventions? Do they pose an issue because they more complex (more septa)? Is there a relationship between frontal sinus size and potential sinusitis—expand on specifically why? The relationship between the maxillary sinus and potential sinusitis has been discussed, owing to the unique drainage patterns.
However, I have to be honest. Even with major re-writes I'm not quite sure how this article really fits into the section it was submitted to: Health Behavior, Chronic Disease and Health Promotion in the International Journal of Environmental Research and Public Health Manuscript. There are many other better suited journals for this type of study.
Author Response
Thank you so much for commenting on my research.
We have corrected contents according to the reviewer comments.
Please see below list and reply if there is any comments.

Reviewer 2 Report
Major flaws -
1. to present the standard values of the frontal sinus in Korean need good number of subjects. Here only 60 CBCT are used.
2. Only gender-based comparisons are done.
3. Conclusion part is totally different from the objectives and results. Growth is not assessed here.
4. The required number of sub- 59 jects was calculated using the G*Power 3.1 program......where is the effect size? Where is a reference?
Author Response

(The authors gave the same response as above.)

Reviewer 3 Report
The authors’ aim was to present the standard values of the frontal sinus in Korean adults by comparing their sizes using a 3D imaging program and to examine the correlation between the dimensions of the frontal sinus and the length of cranial bone.
I may have some remarks:
Please indicate the voxel size of the CBCT scan.
In line 68 revise: “...CT scanning...” IF CBCT scanning was performed, use the wording CBCT scanning
Reference for Table 4 is missing from the text.
Revise the content of sentences of Line 160-162.
I suggest the reduction of the Conclusions section and highlight the conclusions in one or two sentences.
Author Response

(The authors gave the same response as above.)

Round 2
Reviewer 1 Report
The authors have made substantial changes to the manuscript, and they have done a tremendous job addressing my concerns (including the major ones). The incorporation of the boxplots and regression analyses make this study much more robust and meaningful -- it also makes it more unique, in that a lot of studies looking at frontal sinus morphology stick to simpler analyses (such as Pearson correlations). The discussion on power and effect size is an added bonus -- well done! The authors have also done a nice job of making the study more repeatable, by indicated the orientation (Frankfort horizontal) and being more clear about their measures.
The added information about health-related implications make this article now more applicable to the journal's special issue.
I have no other edits-- I am very impressed in the amount of changes the authors put in, especially for the quick turn around.
Author Response
Thank you so much for commenting on my research.
Please let me know if there is anything I need to add
Thank you for your kindness.

Reviewer 2 Report
Major flaws -
1. to present the standard values of the frontal sinus in Korean need a good number of subjects. Here only 60 CBCT are used.
2. Malocclusion-based comparison part added with 10 subjects in each group. Extremely poor sample size.
Author Response

(The authors gave the same response as above.)
